# Wetland vegetation composition and ecology of Lake Abaya in southern Ethiopia

**Dikaso Unbushe Gojamme**[ORCID] *

Biology Department, College of Natural Sciences, Wolaita Sodo University, Wolaita Sodo, Ethiopia

* unbusheg@gmail.com

**Data Availability Statement:** All relevant data are within the manuscript and its Supporting Information files.

**Funding:** The author received no specific funding for this work.

## Abstract

Wetland vegetation and ecology of Lake Abaya in the southern Ethiopia was studied to determine floristic composition, plant community type and vegetation ecology. A total of 102 plots were laid along transects that were set up preferentially across areas where there were rapid changes in vegetation or marked environmental gradients to collect data on estimate of percentage aerial cover of plant species and environmental variables. Vegetation data was analyzed by agglomerative hierarchical cluster analysis using similarity ratio as a resemblance index and Ward's linkage method. Multivariate data analysis was performed using appropriate packages in R version 2.14.0. Canonical Correspondence Analysis (CCA) was used to explore the relationship between the species composition and environmental variables. The environmental data included in the CCA were determined using stepwise backward and forward selection of variables by ANOVA test. Statistical measurement regarding species diversity, richness and evenness of the plant community types was carried out by using Shannon-Wiener diversity indices. A total of 92 plant species belonging to 66 genera and 34 families were identified. Families Poaceae, Asteraceae, Fabaceae, Cyperaceae, Solanaceae, Euphorbiaceae and Amaranthaceae account for about 56.99% of the total proportion. Based on the cluster analysis, five plant community types were identified. The most important factors influencing the plant species composition and pattern of wetland plant communities were water drainage, water depth, land use, slope, altitude, and hydrogeomorphology. Therefore, these factors should be considered in future management and protection under the circumstance of climate change and human activities.

## 1. Introduction

Wetlands are important features in the landscape that provide numerous beneficial services for people and for fish and wildlife. Wetlands are among the most important productive ecosystems on earth because of the complex interactions between biotic (fauna, flora, microbes and unicellular organisms) and abiotic (soil, water and topography) components [1, 2]. Due to the periodic inflow of nutrients, wetland ecosystem is considered to be more productive than the adjacent area. The productivity of wetlands is comparable to rainforests and coral reefs [2]. They constitute a resource of great economic, cultural, scientific and recreational values. [3]

**Competing interests:** The authors have declared that no competing interests exist.

estimated that freshwater wetlands hold more than 40% of the entire world's plant species and 12% of all animal species. [4] estimated that more than 40% of fishes (of the 20,000 species in the world) live in fresh water wetlands. Individual wetlands can be important in supporting high numbers of endemic species; for example, [3] reported that Lake Tanganyika in Central Africa supports 632 endemic fishes and other animal species.

Various physico-chemical and biological characteristics of wetlands regulate cycles of nutrients including C and contribute towards water purification. Depending on biogeochemical processes and hydrology, wetlands could function as net sequesters or producers of greenhouse gases. Although wetlands produce about 40% of the global $CH_4$ emissions, they have the highest C density among terrestrial ecosystems and relatively greater capacities to sequester additional $CO_2$ [5]. Wetlands sequester C through high rates of organic matter inputs and reduced rates of decompositions. Wetlands produce an ecological equilibrium in the environment by maintaining the integrity of life support systems for sustainable socio-economic development. They play important roles in ecological functions in their natural states, which contribute to the well-being of human society.

Human activities cause wetland degradation and loss by changing water quality, quantity, and flow rates; increasing pollutant inputs; and changing species composition as a result of disturbance and the introduction of nonnative species. Even though researchers have paid a great deal of attention to wetland loss and status, the actual extent of wetland loss on a global scale, especially the loss caused directly by human activities, and the actual extent of currently surviving wetlands remains uncertain. By combining datasets related to global wetlands, [6] found that at least 33% of global wetlands had been lost as of 2009, including 4.58 million $km^2$ of non-water wetlands and 2.64 million $km^2$ of open water. In tropical and subtropical areas conversions of wetlands to alternative land uses have accelerated wetland loss and agriculture is considered the principal cause for wetland loss. Wetlands were drained to control disease and for agricultural activities [7–10]. Other important reasons for their vulnerability are the fact that they are dynamic systems undergoing continual change [11] and the fact that they are often open-access resources with limited control over how they are used and what is harvested from them [12].

In Ethiopia, wetlands are distributed across all agro-ecological Zones from high altitude of 4000 m.a.s.l. (meter above sea level to 125 m.b.s.l. (meter below sea level) [13]. The different geological formation, ecological diversity and climatic conditions have endowed Ethiopia with all types of wetlands except coastal, marine-related wetlands and extensive swamp-forest complexes [14].

Ethiopia owns 77 wetland including lakes that cover an area of 13,700 $km^2$, which is about1.14 percent of the country's mainland [15]. Different estimates also indicate that the total area of wetland in Ethiopia may exceed 2% of the country's surface area (22,500 $km^2$) [16]. Wetlands are the main sources of valuable water resources in Ethiopia, where water resources are unevenly distributed and only a quarter of its population has access to safe water. In many parts of Illubabor and Wollega, many perennial and annual springs are associated with the existence of wetlands. Throughout the country, wetlands are important sites for livestock and wildlife grazing especially during the dry period. Floodplains of Borkena, Dabus and Fogera are vital sources of fodder, particularly during dry season, to both domestic and wild animals.

Wetlands are also vital sources of food, fuelwood, and raw materials for making household furniture. A growing number of people in the country, in both rural and urban areas depend on wetland resources for their survival. Poor rural households, particularly women depend on wetlands for additional income to their families. In the ethnic group known as the 'Agnuak' in the Gambella lowlands along banks of the Baro and Gello Rivers, women are heavily involved

with fishing activities [13]. Many peasant farmers in the western parts of the country make their living from wetlands. Sedges are one of the important wetland resources that local communities use in different parts of the country. For example, in western Oromia, sedges have great importance for thatching of houses. More than three fourth of the local households in Illubabor Zone [13], use sedges for roofing their houses.

Wetlands associated with lake Abaya provide range of ecological and economic importance to wildlife and people in the surrounding area. The wetland vegetation is an important nesting and feeding areas for hundreds of wetland birds and hippopotamus and protective shelter for spawning areas for crocodiles. Wetlands of Rift Valley Lakes are sites for tourist attraction.

Despite the global importance, wetlands in Ethiopia are still facing many problems. The major threat comes from the over harvesting of wetland resources, the expansion of human settlements in the main Ethiopian Rift Valley Lakes (Ziway-Shala, Hawassa, Abaya-Chamo and Chew Bahir basins), the construction of dams in Koka and Melka-Wakana, drainage for agriculture in southwest Ethiopia especially in Jimma and Wollega Zones [13]. Wetland resources such as water, fishes and vegetation are subjected to over exploitation. Excessive exploitation of the resources from wetlands can lead to a direct collapse of the wetland and its resources. Disappearance of Lake Haramaya wetlands through excessive use of water for irrigation can be evidence to demonstrate overuse of a resource from wetlands.

Threats to wetlands in Ethiopia also originate from the catchments since wetlands are closely interacting with the catchment. Wetlands in the lowland areas are threatened by encroachment due to shortage of cultivation and grazing areas that results from population pressure in the highlands. Overgrazing by livestock and wildlife leads to loss of biodiversity and compaction of the wetland soil during wet periods which can affect infiltration capacity of the wetland soil. This can in turn affect biodiversity of the wetland. In other instances, catchments are the source of agricultural discharges and will result in increased nutrient load to the wetlands (eutrophication) which leads to the colonization (homogenization) of the habitat by single species which are usually invasive (either exotic or indigenous species). Invasive species are a major threat to global biodiversity and an important cause of biotic homogenization of ecosystems [17]. Invasive species are also threats to wetlands of Ethiopia. Examples of invasive species threatening Ethiopia's wetlands include *Prosopis juliflora* in Awash River basin, *Mimosa pigra* in the Baro-Akobo basin, and water hyacinth, *Eichhornia crassipes*, in Koka reservoirs [18, 19]. Increased accumulation and sedimentation will ultimately accelerate the rate of conversion of the wetland system to a terrestrial one [10]. The low-level awareness of communities regarding the benefits of wetlands, capacity limitations such as lack of skilled manpower, scarcity of wetland focused and coordinated institutions, lack of technical and financial support for wetlands conservation also accelerate loss of wetlands.

Although wetlands are among the most productive ecosystems on earth and protection of threatened natural wetlands and preservation of its biodiversity has received increasing attention globally, wetlands and their resources in Ethiopia are still facing many problems. High population densities within the catchments of the Ethiopian Rift Valley Lakes and in the highlands have been associated with a series of deleterious trends, in particular those arising from the clearance of vegetation for agriculture and overgrazing. Quantification of the magnitude and rate of LULC dynamics of Lake Abaya-Chamo wetland within 1990–2019 showed that wetland area has continuously declined throughout 1990–2000, 2000–2010 and 2010–2019 where its magnitude of shrinkage in the respective periods was 11.4% (700ha), 16% (867 ha) and 31.3% (1,424 ha) [20]. Some of the key challenges to the wetlands of lake Abaya are intensive use of land in the buffer zones for crop production, land loss due to deforestation in the watershed, and eutrophication by nutrients from agricultural fields [21]. Overgrazing of lakeshore vegetation by livestock and overharvesting particular plant species such as

*Aeschynomene elaphroxylon* for the construction of traditional boat, clearance of lakeshore vegetation and trampling due to fishing activities and deforestation in the watershed are also the main threats to the wetlands. The western parts of the two lakes are extensively used for big state farms, which were recently given for private investors.

Despite their importance to maintenance of biodiversity, wetlands and their resources (vegetation in particular) in Ethiopia have been barely investigated and their previous documentation was extremely limited. The vegetation ecology, species composition and diversity of wetlands in Ethiopia have not yet been in the detail it deserved. This is typically reflected in wetlands associated with Lake Abaya in south Ethiopia where research in vegetation ecology, species composition and diversity have not yet been conducted. This is due to capacity limitations such as lack of skilled manpower, finance and neglect by government and researchers. Wetland focused training programmes are very scarce in higher learning institutions of the country. Programmes are not implemented to fill this gap nationally. As a result there is shortage of wetland specialists.

The formulation of sound policy and management strategies on wetlands requires a holistic understanding of how wetland ecosystem functions are affected by natural and anthropogenic [22]. Investigating the floristic composition, structure and diversity status of a wetland could enable to properly manage and sustainably use of those resources. Thus, this study was conducted with the purpose of determining the floristic composition, plant species diversity and richness, plant community types, trends in relationship between plant communities and ecological gradients and to recommend some corrective measures of sustainable management. It is also hoped that this research can provide baseline information for further studies and the issues raised and recommendations given in the study areas will apply to the other wetlands in Ethiopia and neighboring countries in the Horn of Africa.

## 2. Methods and materials

### 2.1. Study area

Lake Abaya is part of the Ethiopian Rift Valley Lakes found in southern Ethiopia (Fig 1). Lake Abaya is the largest lake in the Ethiopian Rift, located between 5˚3'19"N and 6˚45'11"N latitude and 37˚18'55"E and 38˚7'55"E longitude [23]. It has a maximum length of 79.2 km, maximum width of 27.1 km with a surface area of 2600 km2 [21]. It has a maximum depth of 24.5 m and is located at an average altitude of 1,235 m.a.s.l. [21]. The western edges of lake Abaya is covered by alluvial and lacustrine deposits. Fluvisols are the result of alluvial and lacustrine deposits, dominate the Lake Abaya wetland and the adjacent low-lying areas [24]. The fertile luvisols, having good agricultural potential, covered the eastern side of Chamo lake. The soils are intensively used for agricultural production [24]. [25] reported that data generated through in-situ measurements and laboratory analyses of physico-chemical parameters and the plankton communities, when compared with results from the literature, turbidity, salinity, nitrates and soluble reactive phosphorous showed an increasing trend in Lake Abaya. Quantification of the magnitude and rate of LULC dynamics of Lake Abaya-Chamo wetland within 1990–2019 showed that wetland area has continuously declined throughout 1990–2000, 2000–2010 and 2010–2019 where its magnitude of shrinkage in the respective periods was 11.4% (700ha), 16% (867 ha) and 31.3% (1,424 ha) [20].

The Abaya-Chamo lakes basin experiences bimodal rainfall pattern (Fig 2). Lake Abaya-Chamo basin is covered by different vegetation types and experiences different land use practices. The vegetation between the lake and highlands is dominated by shrubs and bush lands. The wetlands associated with Lake Abaya are dominated by tall waterside grasses and leguminous trees, such as *Aeschynomene elaphroxylon*, *Typha angustifolia*, and *Sesbania sesban*. Very

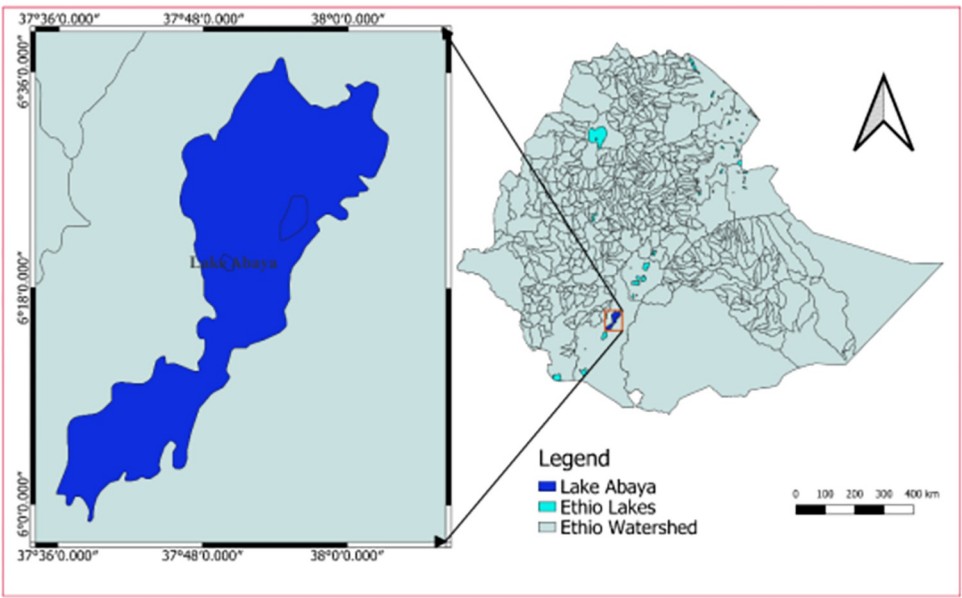

**Fig 1. Map showing location of Lake Abaya.**

dense ground water forest with a number of streams or watersprings covers the shoreline in the southern part of Lake Abaya. The land use of Lake Abaya and Chamo wetlands, especially western side of the two lakes, has been changed rapidly due to extensive deforestation as a result of increased number of populations in the area [21]. The extensive deforestation resulted in the replacement of vegetation cover by cultivated lands. Farming activities like livestock rearing and crop production are the main land use practices in the catchment surrounding the lakes.

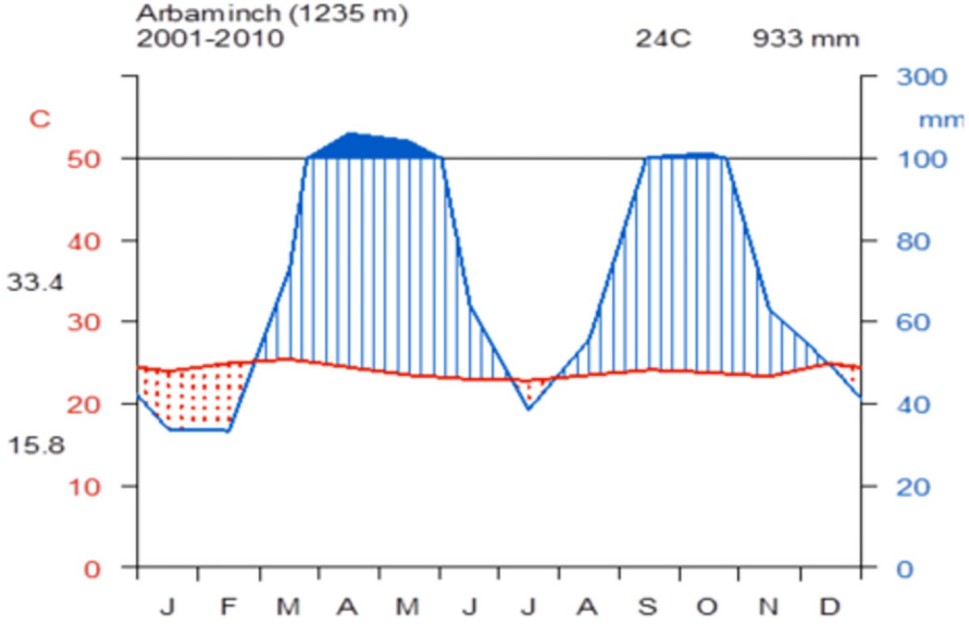

**Fig 2. Climate diagram for the study area.**

## 2.2. Data collection

**2.2.1. Vegetation.** Representative and relatively homogeneous vegetation units of sampling sites were purposively selected on the basis of physiography and physiognomy. Transects were set up preferentially across area where there were rapid changes in vegetation or marked environmental gradients following [26]. Transects were laid out to represent hydrological variations and habitat heterogeneity. Transects were started from the edge of the open water and extended to the ecotone along the edge on both sides of the wetland.

[26, 27] suggested quadrat size for meadow and fen vegetation type is 4 m x 4 m (16 m$^2$) and a shrubby heath, tall herbs with grassland vegetation type are 10 m x 10 m (100 m$^2$). Thus, because of differences in vegetation zonation pattern and continuity of substrates, vegetation data were collected at each site using a plot size of 4 m x 4 m (16 m$^2$) for meadow and fen vegetation type and 10 m x 10 m (100 m$^2$) for shrubby and tall herbs with grassland vegetation type.

A total of 102 sample plots were placed along transects at 50 m intervals in each of the purposively (preferentially) selected habitat units or strata based on moisture gradient and dominant vegetation types. Location of the plots was chosen visually and purposively to ensure that it gives a representative view of the plant species and abundance in that unit. The location of each plot was recorded using a GPS. Within each sample plot in all study areas, all plant species were recorded and the percentage cover of each species was estimated. This was later converted to 1–9 scale following the Braun-Blanquette method modified by [28].

Cover was estimated as the percentage of sampling area covered by the vertical projection of individuals of each species present [26]. Plant species occurring outside the plots were identified to document species diversity of the study area. Voucher specimens were collected, coded, pressed and dried for subsequent identification and verification at the National Herbarium (ETH), Addis Ababa University, using Flora of Ethiopia and Eritrea and those of other neighboring countries.

**2.2.2. Environmental.** The environmental variables recorded in each plot include: drainage, hydrogeomorphology, disturbance, slope and elevation. At each site or sample plot, estimate of disturbance intensity was rated based on physical evidence of the site characteristics such as the presence of soil irregularities, burning, defoliation, trampling and dung.

(Table 1). Plot level hydrogeomorphologic features were rated based on physical evidence of the site or sample plot (e.g., position of the land scape and additional water source) whereas drainage was recorded based on water holding capacity of the soil or level of saturation (Table 1). Depth to water table was measured in each plot with a labeled PVC pipe. Environmental data on topographic parameters such as altitude and coordinates for each plot were determined with GPS.

## 2.3. Data analysis

Multivariate data analysis methods were used to analyze the vegetation and environmental data. Statistical analysis was performed in the R version 2.14 statistical computing program [29] using packages for classification and ordination. Both ordination and classification techniques were employed to study the ecology of wetland vegetation.

Vegetation data were analyzed using agglomerative hierarchical cluster analysis [26] using similarity ratio as a resemblance index and Ward's linkage method to identify vegetation assemblages. Distinct clusters were identified at appropriate hierarchical levels and the quadrats of the data set were then arranged using the sequence of the quadrats in the dendrogram produced. The mean cover value of each species in each cluster identified was calculated and a synoptic table was produced. The species with highest mean cover value was used to determine dominance and sub-dominance of species in cluster groups. Based on the cluster analysis

**Table 1. Environmental variables recorded from the study sites.**

**Hydro-geomorphology**

| Category | Level | Remark |
|---|---|---|
| Open waterbody | 0 | Open waterbody ver sparse acquatic vegetation present |
| Waterlogged or permanently flooded | 1 | Permanently flooded emergent or floating plant found |
| Seasonally flooded | 2 | Seasonally flooded |
| Poorly drained | 3 | Poorly drained area where water may accumulate |
| Well drained | 4 | Low lying landscape with well drained soils (sandy) |
| Wet terrestrial | 5 | Terrestrial landscape with clay soil |
| Dry terrestrial | 6 | Dry terrestrial landscape |

**Drainage**

| Category | level | Remark |
|---|---|---|
| Excessively drained | 1 | low water holding capacity |
| Moderately well drained | 2 | Wet close to surface |
| Poorly drained | 3 | Soils wet to the surface most of the time |
| Standing /flowing water | 4 | Presence of flowing or standing water |

**Slope Category**

| Category | level | Remark |
|---|---|---|
| No Slope/Flat gradient | 0 | Mainly flat at plot location and surrounding landscape |
| Nearly flat | 1 | Nearly flat with no distinct aspect |
| Flat to slight slope | 2 | low gradiant |
| Slight to moderate slope | 3 | Slight to moderate gradiant |
| Moderate to steep slope | 4 | Moderate to steep slope gradiant |
| Steep slope | 5 | Steep slope |

**Disturbance at the local level** (estimate of disturbance intensity based on physical evidence of the site or sample plot characterestics (e.g. soil irregularities, grazing, burning, cultivation, trampling, defoliation and dung) present at the local level).

| Intensity of disturbance | level | Remark |
|---|---|---|
| No disturbance | 0 | No evidence of disturbance |
| Very low disturbance | 1 | Predominantly undisturbed, some physical evidences |
| Low disturbance | 2 | Significant physical evidence |
| Moderate | 3 | Moderate level of physical evidences |
| Highly disturbed | 4 | high level of evidences |
| Very highly disturbed | 5 | Intensively disturbed, Very high level of physical evidences |

output and the resulting synoptic table and ecological evaluation in the field, the number of community types (clusters) was determined at appropriate dissimilarity levels (height of the dendrogram). The plant community types were named after one, two or three dominant species, using the highest mean cover values of plant species which occur in each group. Dominant species are those that are most conspicuous in the community and are high in one or more of the importance values [30], mean cover value in this case.

The identified plant community types were tested for the null hypothesis of no significant difference between the groups using the Multi-response Permutation Procedures (MRPP) [31, 32]. A significance level of 0.05 was used to determine if the compositional differences between groups were statistically significant. The plant community types were examined using nonmetric multidimensional scaling (NMDS) ordination. NMDS was chosen because the tests can produce robust visualizations of data despite numerous zero-values and highly variable data with lack of normality [33]. Using the synoptic table and the habitat information gathered during the sampling period, the different plant communities were described.

Canonical Correspondence Analysis (CCA) [34–37], was used for revealing patterns in the species composition data and relating the patterns to measured environmental variables. CCA technique assumes a unimodal distribution of species in relation to environmental variables [33, 38]. Species cover abundance data for each plot together with the corresponding plot versus environmental variables data matrix were subjected to Canonical Correspondence Analysis (CCA) to reveal the relations between the species composition and environmental variables.

The environmental data included in the CCA were determined using stepwise backward and forward selection of variables by ANOVA test. The CCA generated biplot scores (i.e., correlations between environmental variables and ordination axes) were used to infer the relative importance of each environmental variable for prediction of species composition and distribution [39].

**2.3.1. Diversity analysis.** Statistical measurement regarding species diversity, richness and evenness of the plant community types was carried out by using diversity index [40]. Shannon-Wiener diversity index is universally accepted index for diversity as it accounts for entropy in an ecosystem or in representative samples. It accounts for both species richness and its evenness or equitability. The Shannon-Wiener diversity index (H') was computed using the mean cover values of the species in the plant community types as the input matrix (synopsis output). The Shannon diversity index is calculated by using the formula:

$$H' = -\sum_{i=0}^{s} PilnPi$$

Where $s$ = the number of species; $P_i$ = the proportion of the individuals of the $i^{th}$ species or the abundance of the $i^{th}$ species expressed as a proportion of total cover; Ln = log base$_n$ (natural logarithm). Equitability which determines the relative evenness of the species within the plant community was also calculated. Equitability (Shannon evenness, J) is calculated by the following formula:

$$H' = -\sum_{i=0}^{s} PilnPi$$

$$J = \frac{H'}{Hmax}$$

Evenness index Where, J = the equitability; Hmax = LnS = log base$_n$; S = the number of species

Floristic similarities with regard to species composition among and between study sites were calculated by employing Sorenson's similarity coefficient [27] by using the equation:

$$SC = \frac{2a}{(2a + b + c)}$$

Where SC = Sorenson's similarity coefficient; a = number of species common to both categories; b = number of species present in the first category and absent in the second and c = number of species present in the second category and absent in the first.

# 3. Results

## 3.1. Floristic composition

A total of ninety two plant species belonging to 66 genera and 34 families were recorded from the wetlands of Lake Abaya. Families Poaceae, Asteraceae, Fabaceae, Cyperaceae, Solanaceae,

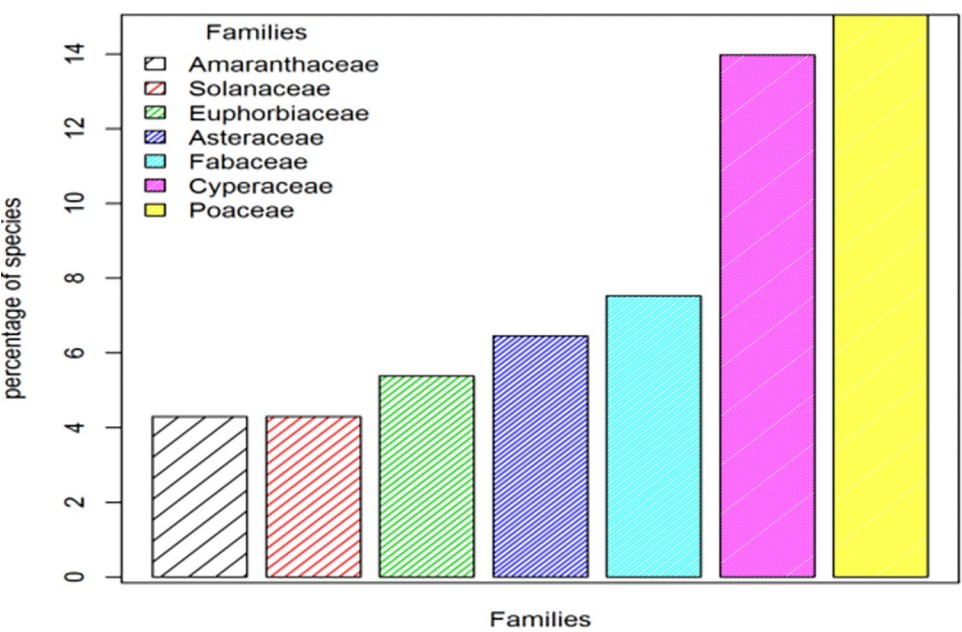

**Fig 3. Families with the greatest representation in Lake Abaya wetland.**

Euphorbiaceae and Amaranthaceae account for about 56.99% of the total proportion (Fig 3). Poaceae was represented by 14 (15.05%) species belonging to 11 genera; Cyperaceae by 13 species (13.98%) belonging to 2 genera; Fabaceae was represented by 7 species belonging to 4 genera while Asteraceae was represented by 6 species belonging to 6 genera. Euphorbiaceae was represented by 5 species belonging to 4 genera; Solanaceae was represented by 4 species belonging to 4 genera and Amaranthaceae was represented by 4 species belonging to 2 genera.

**3.1.1. Description of plant community types of Lake Abaya wetland.** Based on the cluster analysis output and ecological evaluation in the field, five community types were identified between 2.0 to 2.5 heights (0.60–0.75 dissimilarity levels) of the dendrogram (Fig 4). The plant community types were named by the dominant species, which occur in each group, using the highest synoptic values of plant species (Table 2). Communities with their dominant and subdominant species, the number of relevés they contained and diversity index are given in Table 3.

The most common herbaceous species based on mean cover values (Table 2) included *Cynodon aethiopicus, Cyperus articulatus, Cyperus latifolius, Ipomoea aquatica, Leersia hexandra, Eichhornia crassipes, Pistia stratiotes, Cyperus laevigatus, Typha angustifolia* and *Leptochloa fusca*. Species such as *Aeschynomene elaphroxylon, Cynodon aethiopicus, C. articulatus, C. laevigatus, C. latifolius, Cyperus subumbellatus, Eichhornia crassipes, Leersia hexandra, Leptochloa fusca, Pistia stratiotes*, and *Typha angustifolia* were widely and frequently distributed. Each of these species occurred in more than 10% of the quadrats.

**1.** *Cyperus articulatus* **community type:** This community type was dominated by *Cyperus articulatus. Cyperus laevigatus, Typha angustifolia, Cynodon aethiopicus, Aeschynomene elaphroxylon, Leersia hexandra* and *Eichhornia crassipes* were also important species with high mean cover values in the community type. This group included plant species that form wet meadow vegetation. Most of the plants in this community type were emergent macrophytes except few of them were floating plants such as *Nymphaea nouchali* and *Eichhornia crassipes*. Thus, in this community type, emergent plant species were more prominent than free floating,

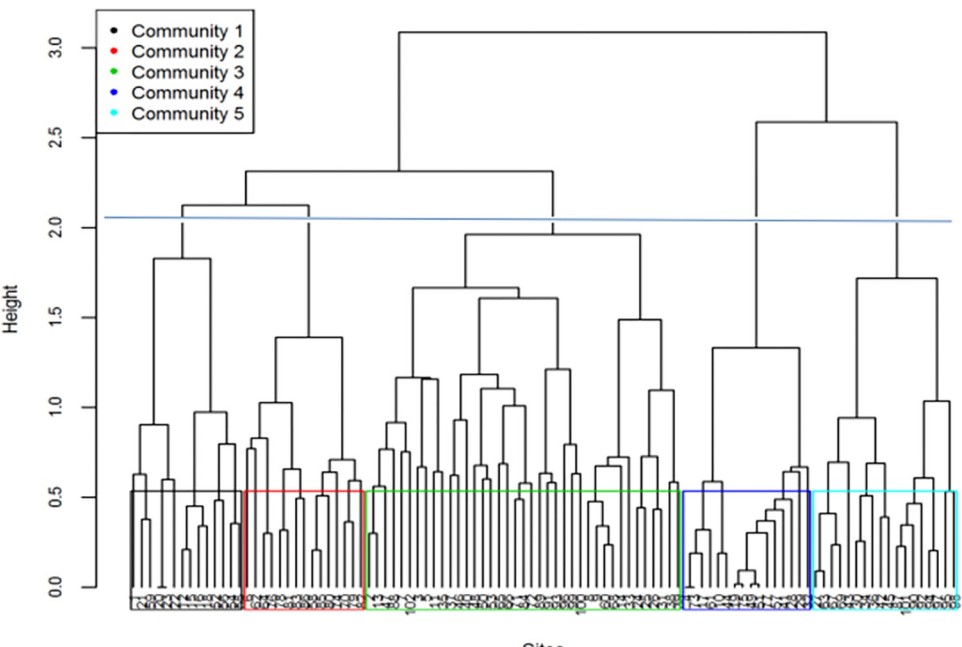

**Fig 4. Dendrogram showing plant community types of Lake Abaya wetland vegetation.**

floating-leaf and submersed plants. This community type consists 13.7% of the stands most of which were from north and northwest of the lakeshore.

**2. *Eichhornia crassipes-Pistia stratiotes* community type:** Dominant species of this community type were *Eichhornia crassipes* and *Pistia stratiotes* (Fig 4). Species such as *Leptochloa fusca*, *Leersia hexandra*, *Cyperus articulatus* and *Aeschynomene elaphroxylon* also contributed high percentage of cover to this community type. This community type was dominated by aquatic and mainly hydrophytic or wetland dependent plant species that are tolerant to high water table. *Eichhornia crassipes*, *Nymphaea lotus*, *Pistia stratiotes*, *Wolffia arrhiza* and *Spirodela polyrrhiza* were free floating and floating-leaf plants in this community type. In this community type, free floating and floating-leaf plants were more prominent than emergent plants, but no submersed plants recorded. These stands were mainly distributed at the edge of the open water bodies.

**3. *Cynodon aethiopicus* community type:** This transitional vegetation type was rich in species and dominanted by *C. aethiopicus* and *Leersia hexandra*. Tree species recorded from this community type were *Ficus sycomorus*, *F. ovata*, *F. sur*, and *Kigelia africana*. Free floating and floating-leaf plants in this community type include *Eichhornia crassipes*, *Nymphaea nouchali*, *Wolffia arrhiza* and *Pistia stratiotes*. This community type was composed of both wetland and upland species and represented species rich community. It had a higher proportion of upland species than the other communities. Thus, free floating and floating-leaf plant species were less prominent than emergent species. All types of floristic composition (forbs, grasses, trees, shrubs and sedges) were observed. This type was distinct from the other wetland communities and showed species richness and composition approaching that of upland vegetation. This community type consisted of 32.5% of the stands which were sampled from south east, southwest and south end of the lake.

**4. *Typha angustifolia-Aeschynomene elaphroxylon* community type:** This community type (Fig 4) was dominanted by *Typha angustifolia* and *Aeschynomene elaphroxylon*. *Leersia*

**Table 2. Synoptic table of Lake Abaya wetland vegetation showing mean cover values of species within clusters.**

| Species | Community types | | | | |
|---|---|---|---|---|---|
| | **C1** | **C2** | **C3** | **C4** | **C5** |
| *Acacia montigena* | 0.000 | 0.000 | 0.308 | 0.000 | 0.000 |
| *Acacia seyal* | 0.000 | 0.000 | 0.846 | 0.000 | 0.000 |
| *Acacia tortilis* | 0.000 | 0.000 | 0.128 | 0.000 | 0.000 |
| *Acalypha fruitcosa* | 0.000 | 0.000 | 0.436 | 0.000 | 0.000 |
| *Acalypha racemosa* | 0.000 | 0.000 | 1.359 | 0.000 | 0.000 |
| *Achyranthes aspera* | 0.000 | 0.000 | 0.795 | 0.000 | 0.000 |
| ***Aeschynomene elaphroxylon*** | 1.286 | 2.667 | 0.179 | **4.750** | **7.111** |
| *Ajuga leucantha* | 0.000 | 0.000 | 0.128 | 0.000 | 0.000 |
| *Aloe otallensis* | 0.000 | 0.000 | 0.308 | 0.000 | 0.000 |
| *Amaranthus dubius* | 0.000 | 0.000 | 0.282 | 0.000 | 0.000 |
| *Amaranthus spinosus* | 0.000 | 0.000 | 0.128 | 0.000 | 0.000 |
| *Phalaris arundinacea* | 0.000 | 0.467 | 0.000 | 0.000 | 0.000 |
| *Balanites aegyptiaca* | 0.000 | 0.467 | 0.589 | 0.000 | 0.000 |
| *Carissa spinarum* | 0.000 | 0.000 | 0.026 | 0.000 | 0.000 |
| *Cayratia ibuensis* | 0.000 | 0.000 | 0.069 | 0.000 | 0.000 |
| *Cissus quadrangularis* | 0.000 | 0.000 | 0.385 | 0.000 | 0.000 |
| *Clausena anisata* | 0.000 | 0.000 | 0.051 | 0.000 | 0.000 |
| *Commelina diffusa* | 0.000 | 0.000 | 0.103 | 0.000 | 0.000 |
| *Cordia africana* | 0.000 | 0.000 | 0.359 | 0.000 | 0.000 |
| *Croton macrostachyus* | 0.000 | 0.000 | 0.564 | 0.000 | 0.000 |
| ***Cynodon aethiopicus*** | 1.642 | 2.533 | **3.564** | 0.000 | 1.722 |
| *Cyperus alopecuroides* | 0.000 | 0.400 | 0.410 | 0.375 | 0.000 |
| ***Cyperus articulatus*** | **4.714** | 1.933 | 0.154 | 0.000 | 1.111 |
| *Cyperus digitatus* | 0.000 | 0.000 | 0.128 | 0.000 | 0.000 |
| *Cyperus distans* | 0.000 | 0.000 | 0.154 | 0.000 | 0.000 |
| *Cyperus grandibulbosus* | 0.429 | 0.000 | 0.000 | 0.000 | 0.000 |
| *Cyperus laevigatus* | 2.357 | 0.267 | 0.282 | 0.000 | 0.000 |
| *Cyperus latifolius* | 0.000 | 3.133 | 0.000 | 0.000 | 0.000 |
| *Cyperus papyrus* | 0.000 | 0.000 | 0.000 | 0.438 | 0.000 |
| *Cyperus rotundus* | 0.000 | 0.000 | 0.179 | 0.000 | 0.000 |
| *Cyperus subumbellatus* | 0.000 | 0.000 | 0.436 | 0.000 | 2.444 |
| *Echinochloa haploclada* | 0.500 | 0.000 | 0.000 | 0.000 | 0.000 |
| *Echinochloa pyramidalis* | 0.000 | 0.000 | 0.436 | 0.625 | 1.333 |
| *Eclipta prostrata* | 0.000 | 0.333 | 0.026 | 0.000 | 0.000 |
| ***Eichhornia crassipes*** | 0.923 | **5.433** | 0.154 | 0.000 | 1.222 |
| *Enneapogon desvauxii* | 0.357 | 0.000 | 0.385 | 0.000 | 0.000 |
| *Eragrostis japonica* | 0.000 | 0.000 | 0.154 | 0.375 | 0.000 |
| *Eriochloa fatmensis* | 0.000 | 0.000 | 0.000 | 0.250 | 0.000 |
| *Euphorbia tirucalli* | 0.000 | 0.000 | 0.154 | 0.000 | 0.000 |
| *Ficus ovata* | 0.000 | 0.000 | 0.538 | 0.000 | 0.000 |
| *Ficus sur* | 0.000 | 0.000 | 0.308 | 0.000 | 0.000 |
| *Ficus sycomorus* | 0.000 | 0.000 | 1.000 | 0.438 | 0.000 |
| *Geigeria alata* | 0.000 | 0.000 | 0.231 | 0.000 | 0.000 |
| *Gossypium barbadense* | 0.000 | 0.000 | 0.179 | 0.000 | 0.000 |
| *Grewia ferruginea* | 0.000 | 0.000 | 0.128 | 0.000 | 0.000 |
| *Grewia villosa* | 0.000 | 0.000 | 0.000 | 0.000 | 0.278 |

*(Continued)*

**Table 2.** (Continued)

| Species | Community types | | | | |
|---|---|---|---|---|---|
| | C1 | C2 | C3 | C4 | C5 |
| *Hippocratea africana* | 0.000 | 0.000 | 0.154 | 0.000 | 0.000 |
| *Hypoestes forskaolii* | 0.000 | 0.000 | 0.231 | 0.000 | 0.000 |
| *Ipomoea aquatica* | 0.000 | 0.133 | 0.000 | 0.500 | 0.556 |
| *Ipomoea eriocarpa* | 0.000 | 0.000 | 0.077 | 0.563 | 0.167 |
| *Kigelia africana* | 0.000 | 0.000 | 0.179 | 0.000 | 0.000 |
| ***Leersia hexandra*** | 2.643 | 2.667 | 2.154 | 0.688 | 0.556 |
| *Leptadenia arborea* | 0.000 | 0.000 | 0.051 | 0.000 | 0.000 |
| *Leptochloa fusca* | 0.357 | 2.067 | 0.026 | 0.563 | 1.556 |
| *Leucas deflexa* | 0.000 | 0.000 | 0.179 | 0.000 | 0.000 |
| *Ludwigia stolonifera* | 0.000 | 0.000 | 0.103 | 0.000 | 0.000 |
| *Mangifera indica* | 0.000 | 0.000 | 0.051 | 0.000 | 0.000 |
| *Maytenus arbutifolia* | 0.000 | 0.333 | 0.154 | 0.000 | 0.000 |
| *Maytenus senegalensis* | 0.000 | 0.000 | 0.538 | 0.000 | 0.111 |
| *Melhania ovata* | 0.000 | 0.200 | 0.000 | 0.000 | 0.000 |
| *Nymphaea lotus* | 0.000 | 0.600 | 0.000 | 0.000 | 0.000 |
| *Nymphaea nouchali* | 0.429 | 0.000 | 0.128 | 0.000 | 1.556 |
| *Phragmates karka* | 0.000 | 0.000 | 0.000 | 0.313 | 0.000 |
| ***Pistia stratiotes*** | 0.000 | **4.533** | 0.128 | 0.000 | 0.389 |
| *Plecthrantus barbatus* | 0.000 | 0.000 | 0.128 | 0.000 | 0.000 |
| *Pluchea ovalis* | 0.000 | 0.000 | 0.872 | 0.000 | 0.000 |
| *Potamogeton pucillus* | 0.000 | 0.200 | 0.000 | 0.125 | 0.000 |
| *Pulcaria schimperi* | 0.000 | 0.000 | 0.077 | 0.000 | 0.000 |
| *Senna didymobotrya* | 0.000 | 0.000 | 0.359 | 0.000 | 0.333 |
| *Senna septemtrionalis* | 0.000 | 0.000 | 0.000 | 0.000 | 0.278 |
| *Sesbania sesban* | 0.000 | 0.067 | 0.179 | 0.000 | 0.000 |
| *Solanum incanum* | 0.000 | 0.000 | 0.949 | 0.000 | 0.000 |
| *Sorghum verticilliflorum* | 0.357 | 0.333 | 0.205 | 0.813 | 0.000 |
| *Spirodela polyrrhiza* | 0.000 | 0.000 | 0.026 | 0.000 | 0.222 |
| *Trichilia emetica* | 0.000 | 0.000 | 0.231 | 0.000 | 0.000 |
| ***Typha angustifolia*** | 2.471 | 0.400 | 0.308 | **7.875** | 0.000 |
| *Withania somnifera* | 0.000 | 0.000 | 0.077 | 0.000 | 0.000 |
| *Wolffia arrhiza* | 0.000 | 0.533 | 0.179 | 0.000 | 0.000 |
| *Xanthium strumarium* | 0.000 | 0.000 | 0.615 | 0.063 | 0.333 |

*hexandra* was another important species contributing most to the composition of this community type. *Ficus sycomorus* and *Aeschynomene elaphroxylon* were tree species in this community type. This community type consisted of 15.7% of the stands which were from north, northwest, and west of the lake. All of the plants in this community type were emergent macrophytes except *Potamogeton pucillus*, a submerged aquatic plant. In this community type, free floating and floating-leaf plants were not recorded. Emergent plants were more prominent than free floating, floating-leaf and submersed plants.

**5. *Aeschynomene elaphroxylon* community type:** This shruby marsh community type was dominanted by *Aeschynomene elaphroxylon* (Fig 4). *Nymphaea nouchali*, *Echinochloa pyramidalis*, *Leptochloa fusca*, *Eichhornia crassipes* and *Cyperus articulatus were* sub dominant species in this community type. This community type was represented by 17.6% of the stands sampled from north, northwest, west, and northeast of the lake. Free floating and floating-leaf plants

**Table 3. Summary of wetland plant community types from Lake Abaya vegetation data.**

| Agglomerative hierarchical relationships | Plant community types | | | | |
|---|---|---|---|---|---|
| | **1** | **2** | **3** | **4** | **5** |
| **Dominant species** | *Cyperus articulatus* | *E. crassipes* *P. stratiotes* | *Cynodon aethiopicus* | *T. angustifolia A. elaphroxylon* | *Aeschynomene elaphroxylon* |
| **Mean cover values** | 4.71 and 3.36 | 5.43 and 4.53 | 3.56 | 7.88 and 4.75 | 7.11 |
| **Sub-dominant species** | *T. angustifolia, C. aethiopicus, A. elaphroxylon* and *L. hexandra* | *L. fusca, L. hexandra, C. articulatus, C. aethiopicus,* and *A.elaphroxylon* | *Leersia hexandra, F. sycomorus* and *A. fruitcosa* | *Leersia hexandra* | *C. aethiopicus, N. nouchalii, E. pyramidalis, L. fusca, E. crassipes, C. articulatus* |
| **Shannon's diversity x(H)** | 2.10 | 2.56 | **3.64** | 1.87 | 2.31 |
| **Shannon's evenness (J)** | 0.84 | 0.82 | **0.87** | 0.67 | 0.80 |
| **Richness (S)** | 12 | 23 | **67** | 16 | 18 |
| **No. quadrats** | 14 | 15 | **39** | 16 | 18 |

recorded from this community type were (*Nymphaea nouchali, Pistia stratiotes, Spirodela polyrrhiza* and *Eichhornia crassipes*). Similar to community types two and four, this community type occurred on the edges of open and deeper standing water.

**3.1.2. Diversity of plant community types.** The Shannon-Wiener Diversity computed for five different plant communities (Table 3) showed that community type three (*Cynodon aethiopicus*) which was found in relatively dry terrestrial landscape and exposed for disturbance was the most diverse and has the highest species richness and evenness. Community type four (*Typha angustifolia-Aeschynomene elaphroxylon*) which occurred in deeper standing water around the open water zones and experiencing prolonged inundation had the least diversity index and had the least even distribution of species. Community type one (*Cyperus articulatus*) was the least in species richness and occurred in deeper water next to community four.

## 3.2. Environmental gradients influencing plant communities

**3.2.1. Multivariate Non-metric Multidimensional Scaling.** Multivariate Non-metric Multidimensional Scaling (NMS) (McCune and Grace 2002) resulted in a 2-dimensional solution with a final stress of 0.255 (Fig 5). The first axis was most correlated with water depth ($r^2$ = 0.248, p = 0.001), Hydrogeomorphology ($r^2$ = 0.167, p = 0.003) and slope ($r^2$ = 0.334, p = 0.001) (Table 3). Community type five (*Aeschynomene elaphroxylon*) was mostly correlated with drainage ($r^2$ = 0.067, p = 0.039) where as community type three was highly correlated with slope $r^2$ = 0.334, p = 0.001) (Table 4). *Typha angustifolia-Aeschynomene elaphroxylon* (community four) which mainly consisted of wetland dependent species was correlated with higher water depth ($r^2$ = 0.248, p = 0.001) (Table 4).

**3.2.2. Canonical correspondence analysis.** Species and plots ordination in the space defined by the first two CCA axes are shown in Fig 6A and 6B. CCA ordination illustrates the relationship of the four environmental variables (water depth, drainage, slope and hydrogeomorphology) to five community types and to species distribution pattern. The first two axes together accounted for 63% (Table 5) of the variation explained. The proportion of variances explained by these axes was 36% and 27%, respectively (Table 5).

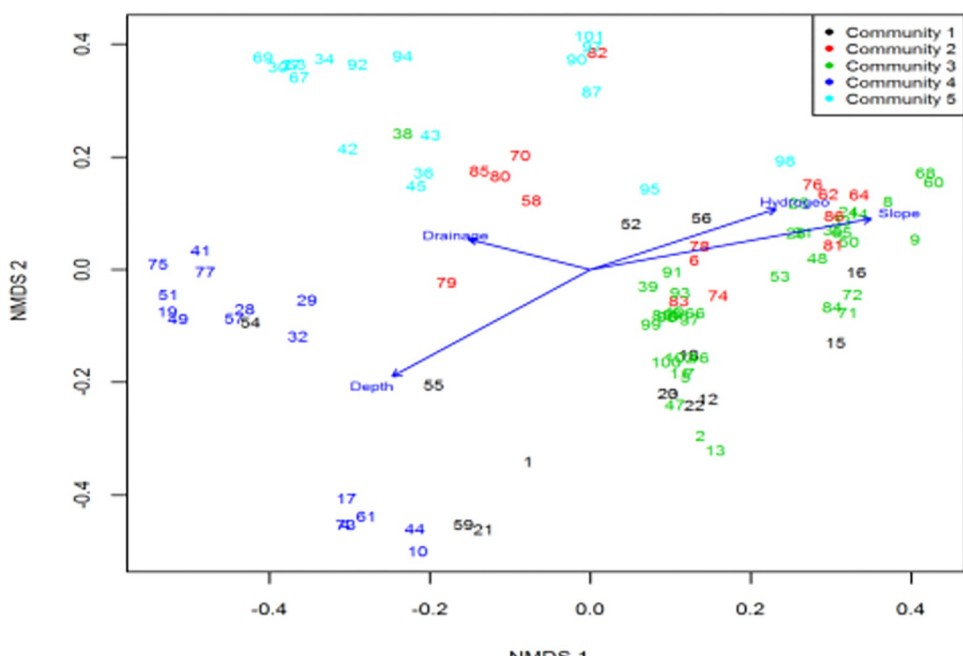

**Fig 5. Non-metric Multidimensional Scaling (NMDS) diagram of plots of Lake Abaya vegetation grouped by community type (communities 1 to 5).** Vectors represent environmental variables: depth, hydrogeomorphology, drainage and slope.

Results from one way ANOVA test showed that water depth, slope, drainage and hydrogeo-morphology were significantly related to the floristic composition of the plant community types. The first axis extracted by the analysis was closely related to the drainage and slope, as indicated by the biplot scores of -0.43 and -0.78 (Table 5). The second axis extracted by the analysis was more closely related to water depth and hydrogeomorphology, as indicated by the biplot scores of -0.71 and 0.54, respectively (Table 5). The second axis was associated negatively with water depth and positively with hydrogeomorphology. Canonical coefficients from the best-fit multiple regression models (ANOVA test), biplot scores for the constraining variables, eigenvalues and proportion of variances explained by the first two axes were indicated. Factor values that differed significantly between the groups according to an ANOVA are shown in bold.

Ordination diagram of CCA (Fig 6A) for plots and environmental factors showed that *Eichhornia crassipes-Pistia stratiotes* (community two) and *Typha angustifolia-Aeschynomene*

**Table 4. Pearson ($r^2$) correlations of Nonmetric Multidimensional Scaling ordination axes with environmental variables of Lake Abaya vegetation.**

| Vectors | NMDS1 | NMDS2 | $r^2$ | Pr(>r) |
|---|---|---|---|---|
| Depth | -0.793 | -0.609 | 0.248 | 0.001*** |
| Drainage | -0.942 | 0.335 | 0.067 | 0.039 |
| Hydrogeo | 0.906 | 0.423 | 0.167 | 0.001*** |
| Slope | 0.968 | 0.250 | 0.334 | 0.001*** |

Significance codes: 0 '***' 0.001 '**' 0.01 '*' 0.05 '.' 0.1 ' ' 1

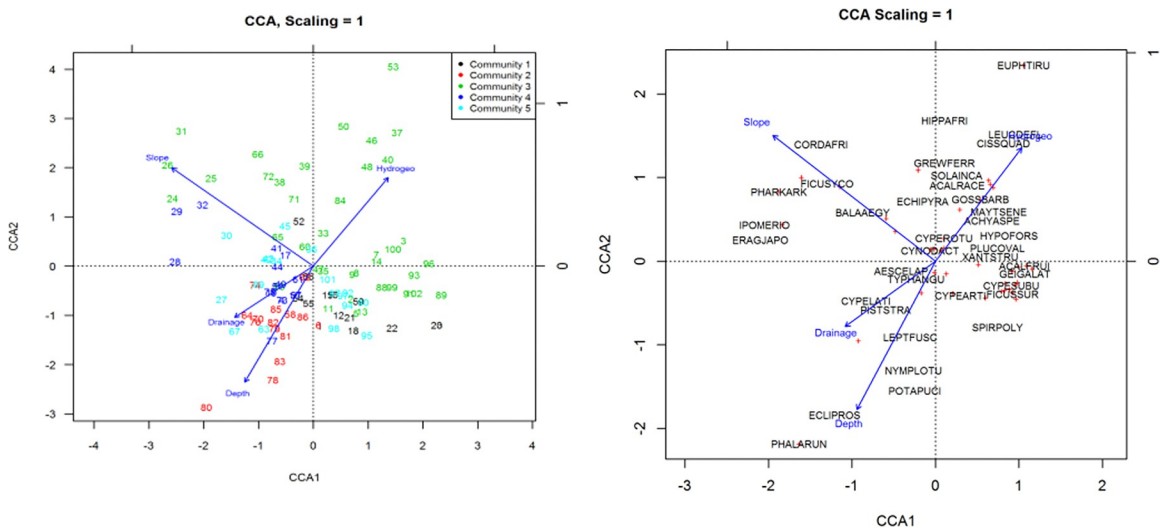

**Fig 6. Canonical correspondence analysis (CCA) of wetlands of Lake Abaya vegetation.** (a): shows the relationships between the plant community types and environmental variables, (b): shows the relationships between plant species distribution and environmental variables. Only higher priority species with high variances are visible and all other less dominant species are indicated by plus sign (+). Species were abbreviated by combining the first four letters from generic names and specific epithets. Environmental attributes represented by the vectors included water depth, slope, drainage and hydrogeomorphology. The vectors representing environmental gradients point in the direction of the most rapid change in each variable.

*elaphroxylon* (community four) were found in areas with high water level and poor drainage. Therefore, distribution of these community types was significantly affected by hydrologic factors such as water depth, drainage and hydrogeomorphology. These community types were consisted of wetland dependent plant species that are tolerant to high water table which most commonly occur in waterlogged areas characterized by presence of standing or permanently flowing water for most of the year. This wetland dependent community type also occurred in permanently flooded areas as well as in and around the edge of open water bodies.

Alternatively, *Cynodon aethiopicus* (community type 3) with large number of plots (in the upper part of the ordination space (Fig 6B) was found in excessively drained areas with high disturbance and low water level. These groups also commonly occurred in areas characterized by low lying dry terrestrial landscape with well drained soils. Water depth and hydrogeomorphology were strong factors differentiating these groups from other community types. Slope was a strong factor differentiating community type five (*Aeschynomene elaphroxylon*) from others. It was relatively intact and pristine or semi-natural vegetation found in areas occasionally used for grazing activities.

Analysis of variance showed that there was a strong relationship between species and environmental factors such as water depth ($p<0.01$), drainage ($p<0.01$), hydrogeomorphology ($p<0.05$) and slope ($p<0.01$). The species and environment correlations with the first axis was 0.85 and the second axis was 0.74 (Table 4). Canonical correspondence analysis (CCA) (Fig 6B) for the species and explanatory variables revealed essentially the same pattern as ordination of plots and explanatory variables.

The analysis of variance also showed that there was a marked relationship between the environment and the scores of many species. The plant species *Pharagmatis karka*, *Cordia africana*, *Ficus sycomorus* and *Eragrostis japonica* had strong relationships with the first axis (more closely related to slope). *Euphorbia tirucalli*, *Hippocratea africana* and *Leucas deflexa* were more closely related to areas with decreasing water depth and drainage), whereas *Phalaris*

**Table 5. Results of CCA analysis of Lake Abaya wetland vegetation data.**

| Variables | CCA1 | CCA2 | SSqs | MSqs | F | R2 | Pr(>F) |
|---|---|---|---|---|---|---|---|
| Slope | **-0.78** | 0.60 | 1.32 | 1.32 | 3.78 | 0.03 | 0.01** |
| Depth | -0.38 | **-0.71** | 2.13 | 2.13 | 6.10 | 0.05 | 0.01** |
| Hydrogeomorphlogy | 0.41 | **0.54** | 0.71 | 0.71 | 2.02 | 0.02 | 0.05* |
| Drainage | **-0.43** | -0.31 | 0.97 | 0.97 | 2.78 | 0.02 | 0.01** |
| Eigenvalues | 0.41 | 0.31 | | | | | |
| % variance explained | 0.36 | 0.27 | | | | | |
| Cumulative % variance | 0.36 | 0.63 | | | | | |
| Species enviroment correlations | 0.85 | 0.74 | | | | | |

(* = 0.01 < p < 0.05

** = p < 0.01 and

*** = p < 0.001).

*arundinacea*, *Eichhornia crassipes*, *Pistia stratiotes*, *Nymphaea lotus* and *Potamogeton pucillus* had significant relationship with increasing water depth and drainage.

## 4. Discussion

The most important process that is thought to play a vital role in determining the habitat diversity of the wetland which in turn affect plant community development in area is environmental heterogeneity. Microtopographic variation may also contribute to the diversity of wetland plant communities in wetlands of Lake Abaya. [41] pointed out that changes in the environment due to change in land-use type, drainage, eutrophication or overgrazing can alter spatial heterogeneity in plant community composition.

The most extensive and distinctive plant community in wetlands of Lake Abaya was *Aeschynomene elaphroxylon* and *Typha angustifolia*. The presence of *Eichhornia crassipes-Pistia stratiotes* community in the wetlands of Lake Abaya reveals strong human activities in the study area.

Variation in plant distribution, abundance, and floristic composition of wetland plant communities could be associated primarily with the variation in geographic location, hydrologic condition, patterns and degree of disturbance, various catchment land uses in the surrounding areas of the wetlands. Comparing the ordination and classification results of the plant communities, both methods displayed hydrologic gradients (water depth and hydro geomorphology), environmental factors (elevation and slope) and factors related to anthropogenic activities (drainage and disturbance) were important factors for plant species composition and vegetation assembly.

Results of canonical correspondence analyses and one way ANOVA test showed that factors such as water depth, slope, drainage and hydrogeomorphology were significant factors that differentiated the floristic composition and distribution of the plant community types in Lake Abaya wetlands.

Plant community type three (*Cynodon aethiopicus*) which was found in relatively drier terrestrial landscape and in areas exposed for moderate disturbance was the most diverse and had the highest species richness and evenness values. All types of floristic composition (forbs, grasses, trees, shrubs and sedges) were observed. In contrast to this, plant community types which occurred in deeper or standing water and experiencing prolonged inundation (*Cyperus articulatus*, *Typha angustifolia* and *Aeschynomene elaphroxylon* were the least in species richness. Plant community types recorded from such areas were mainly dominated by few species.

This is consistent with the proposal that permanent inundation may produce water zones dominated by stands of few species and a loss of terrestrial taxa [42]. This also agrees with those of [43]) who found that vegetation in frequently flooded and thus often waterlogged areas were characterized by high aerial biomass and low species diversity. The presence of standing water also affects the ability of some wetland plants to germinate leading to limited recruitment of new species in those zones and a subsequent reduction in species richness [44]. Low species diversity of open water zones may be related to homogeneity of the aquatic habitats compared with the terrestrial ones. Moreover, the low diversity of the open water zone may be because most of its species are highly specific to that aquatic habitat, thus the same species occurs at nearly all sites. Research in different wetlands had indicated that prolonged inundation leads to low plant diversity [45]. [13] emphasized that water depth is frequently the most important environmental factor affecting plant species composition and distribution in wetlands.

Some plant species from most community types occupied at around the origin of the CCA ordination diagram generated from the vegetation data show that they prefer to grow in habitats of moderate values of all the parameters measured. Plant community type one was less distinctly recognized in the CCA ordination diagram due to the presence of species that may tolerate a wider range of habitat conditions. This further suggests that no single environmental factor but rather a combination of factors were significant in defining the variability within the species data. The species ordination diagram displayed a measure of overlap among vegetation types. One of the insights obtained from the ordination process was that the dominant graminoids (with several exceptions) were rather broadly distributed and overlapping in ordination space.

Furthermore, the study on the wetlands resulted in plant communities of emergent species, aggregating free floating, floating leaf and submerged plants as minor members of associations dominated by the more obvious emergent species and did not indicate the differences that may exist between emergent and submerged plants as distinct components of wetland communities. This could be due to shared affinities for environmental conditions.

## 5. Conclusion and recommendations

The study was resulted in the documentation of 92 plant species representing 66 genera and 34 families. Most of the plants in the wetlands under investigation were emergent macrophytes except a few of them that were free-floating, floating-leaf and submerged species. The distribution and composition of plant species and vegetation assembly at study site were influenced significantly by a combination of different environmental factors. Variation in plant distribution, abundance, and floristic composition of wetland plant communities could be associated primarily with the variation in geographic location, hydrologic condition, patterns and degree of disturbance, various catchment land uses in the surrounding areas of the wetlands. In the wetland, the dominant plant species were widely distributed resulting in an overlap among plant community types.

Results of the study showed that there were differences in the spatial patterns of plant species composition and diversity due to variations in environmental gradients (hydrologic factors in particular) and anthropogenic factors. Some plant community types were less distinctly recognized in ordination analysis due to the presence of species that may tolerate a wider range of habitat conditions. This further suggests that no single environmental factor but rather a combination of factors were significant in defining the variability within the species data. Therefore, this should be considered in future management and protection under the circumstance of human activities. Overall, results support the prediction that different

environmental variables had varying influence on the overall plant species composition, diversity and distribution.

In order to safeguard the study area from destructive anthropogenic impacts, protective measures should be developed. If proper management strategies not taken timely, the damage can become irreversible. The effective and continuous protection measures should be identified and local community should be informed about the importance of the conservation of these fragile ecosystems. Wetland managers and decisions makers at all levels should consider this baseline data on the species composition and vegetation ecology to guide management decisions and to detect changes over time and space as a result of management and impacts of anthropogenic alteration.

Further study of soil nutrients and water chemistry data is recommended to better understand the compositional variation found between and among the wetlands. It is also recommended that an assessment of the vegetation in terms of its variability and reaction to various practices such as burning and grazing should be done.

## Supporting information

**S1 Data.**
(XLSX)

## Acknowledgments

The author would like to acknowledge the National Meteorological Agency for providing free temperature and rainfall data. The staff members of the National Herbarium, Ethiopia, are also greatly acknowledged.

## Author Contributions

**Conceptualization:** Dikaso Unbushe Gojamme.

**Data curation:** Dikaso Unbushe Gojamme.

**Formal analysis:** Dikaso Unbushe Gojamme.

**Investigation:** Dikaso Unbushe Gojamme.

**Methodology:** Dikaso Unbushe Gojamme.

**Validation:** Dikaso Unbushe Gojamme.

**Writing – original draft:** Dikaso Unbushe Gojamme.

**Writing – review & editing:** Dikaso Unbushe Gojamme.

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
