## [Decision Letter · Decision Letter 0]

15 Aug 2023

PONE-D-23-20697Wetland Vegetation Composition and Ecology of Lake Abaya in Southern EthiopiaPLOS ONE

Dear Dr. Gojamme,

Thank you for submitting your manuscript to PLOS ONE. After careful consideration, we feel that it has merit but does not fully meet PLOS ONE’s publication criteria as it currently stands. Therefore, we invite you to submit a revised version of the manuscript that addresses the points raised during the review process.

We look forward to receiving your revised manuscript.

Kind regards,

Godwin Upoki Anywar, BSc, Msc, PhD

Academic Editor

PLOS ONE

Journal Requirements:

2. Thank you for submitting the above manuscript to PLOS ONE. During our internal evaluation of the manuscript, we found significant text overlap between your submission and previous work in the [introduction, conclusion, etc.].

Please revise the manuscript to rephrase the duplicated text, cite your sources, and provide details as to how the current manuscript advances on previous work. Please note that further consideration is dependent on the submission of a manuscript that addresses these concerns about the overlap in text with published work.

[If the overlap is with the authors’ own works: Moreover, upon submission, authors must confirm that the manuscript, or any related manuscript, is not currently under consideration or accepted elsewhere. If related work has been submitted to PLOS ONE or elsewhere, authors must include a copy with the submitted article. Reviewers will be asked to comment on the overlap between related submissions (http://journals.plos.org/plosone/s/submission-guidelines#loc-related-manuscripts).]

We will carefully review your manuscript upon resubmission and further consideration of the manuscript is dependent on the text overlap being addressed in full. Please ensure that your revision is thorough as failure to address the concerns to our satisfaction may result in your submission not being considered further.

"NO - Include this sentence at the end of your statement: The funders had no role in study design, data collection and analysis, decision to publish, or preparation of the manuscript."

5. We note that [Figure 1] in your submission contain [map/satellite] images which may be copyrighted. All PLOS content is published under the Creative Commons Attribution License (CC BY 4.0), which means that the manuscript, images, and Supporting Information files will be freely available online, and any third party is permitted to access, download, copy, distribute, and use these materials in any way, even commercially, with proper attribution. For these reasons, we cannot publish previously copyrighted maps or satellite images created using proprietary data, such as Google software (Google Maps, Street View, and Earth). For more information, see our copyright guidelines: http://journals.plos.org/plosone/s/licenses-and-copyright.

Reviewers' comments:

Reviewer's Responses to Questions

**Comments to the Author**

1. Is the manuscript technically sound, and do the data support the conclusions?

Reviewer #1: Partly

Reviewer #2: Partly

2. Has the statistical analysis been performed appropriately and rigorously? 

Reviewer #1: Yes

Reviewer #2: No

3. Have the authors made all data underlying the findings in their manuscript fully available?

Reviewer #1: Yes

Reviewer #2: No

4. Is the manuscript presented in an intelligible fashion and written in standard English?

Reviewer #1: Yes

Reviewer #2: No

5. Review Comments to the Author

Reviewer #1: 1. The manuscript is generally well written with adequate statistical analysis. However, the discussion section is thin with some results not discussed at all.

2. The author needs to state how sampling effort was measured.

3. The citation of references is good but majority are quite old considering the dynamism in wetlands loss/degradation.

4. The recommendations and conclusions do not clearly originate from the results perhaps due to the thin discussion. In fact some appear to be from general knowledge.

5. Most of the comments are included in the manuscript as comments

Reviewer #2: To the Editor and the author

Introduction: The often overlooked functions of wetlands, such as their role in purification and carbon sequestration, can diminish the perceived significance of these ecosystems and consequently reduce the motivation to conserve them, particularly within the perspective of policy makers.

Study Area: The graphical representation of the study area depicted in Figure 1 raises concerns due to its poor quality. The aspect ratio of latitude and longitude coordinates is not accurately represented, necessitating a revision of the map to ensure an appropriate scale and resolution. Moreover, the climadiagram's horizontal axis exhibits an exaggeration that may distort its true representation.

Data Collection: The explanation provided for the methods employed in data collection lacks comprehensive detail. A more in-depth description could draw insights from the approach employed in measuring drainage and slope, aiding in establishing a clearer understanding of the study's data gathering procedures.

Data Analysis: The utilization of an outdated version of R during the data analysis, when compared to the more recent version 4.3.1, raises concerns. Employing an outdated version has the potential to influence the outcomes of data analysis, as is evident in the substandard quality of the dendrogram presented in Figure 4. Enhancements such as the incorporation of colored leaves (branches) and labels associated with the required number of clusters can considerably improve the dendrogram's effectiveness. While the dendrogram demonstrates differences in clusters, the rationale behind selecting cluster identification heights spanning the dendrogram's various levels is provided, even when the dissimilarity between the clusters is significant. A decision to recognize just four distinct clusters arises when the cutoff height falls between 2 and 2.5.

The application of NMDS (Non-Metric Multidimensional Scaling) facilitates the correlation of environmental factors with the clusters that have been identified. In cases where NMDS has already produced satisfactory outcomes, the relevance of employing CCA (Canonical Correspondence Analysis) in establishing the relationship between sample plots and environmental factors requires a compelling justification.

What is the significance of the red cross symbols within Figure 6, representing the Canonical Correspondence Analysis (CCA) involving species and environmental factors? Can these symbols be eliminated if needed?

6. PLOS authors have the option to publish the peer review history of their article (what does this mean?). If published, this will include your full peer review and any attached files.

Reviewer #1: No

Reviewer #2: No

---

## [Author Response · Author response to Decision Letter 0]

23 Sep 2023

I have attached response to reviewers and editors' comments as a file.

---

## [Decision Letter · Decision Letter 1]

7 Feb 2024

PONE-D-23-20697R1Wetland vegetation composition and ecology of Lake Abaya in southern Ethiopia.PLOS ONE

Dear Dr. Gojamme,

Thank you for submitting your manuscript to PLOS ONE. After careful consideration, we feel that it has merit but does not fully meet PLOS ONE’s publication criteria as it currently stands. Therefore, we invite you to submit a revised version of the manuscript that addresses the points raised during the review process.

We look forward to receiving your revised manuscript.

Kind regards,

Dharmendra Kumar Meena

Academic Editor

PLOS ONE

Additional Editor Comments:

article is recommended for major revisions

Reviewers' comments:

Reviewer's Responses to Questions

**Comments to the Author**

1. If the authors have adequately addressed your comments raised in a previous round of review and you feel that this manuscript is now acceptable for publication, you may indicate that here to bypass the “Comments to the Author” section, enter your conflict of interest statement in the “Confidential to Editor” section, and submit your "Accept" recommendation.

Reviewer #2: All comments have been addressed

2. Is the manuscript technically sound, and do the data support the conclusions?

Reviewer #2: Yes

3. Has the statistical analysis been performed appropriately and rigorously? 

Reviewer #2: Yes

4. Have the authors made all data underlying the findings in their manuscript fully available?

Reviewer #2: Yes

5. Is the manuscript presented in an intelligible fashion and written in standard English?

Reviewer #2: Yes

6. Review Comments to the Author

Reviewer #2: The manuscript has improved significantly, but the dendrogram and the CCA figures are a bit blur. Saving the output of the analyses in publication quality could help in improving the clarity of the figures. The identification of the clusters could be done in a much presentable manner than identifying them surrounding them with rectangles. learning how to do that could be useful for future works as well.

7. PLOS authors have the option to publish the peer review history of their article (what does this mean?). If published, this will include your full peer review and any attached files.

Reviewer #2: **Yes: **Zerihun Woldu Tesema

---

## [Author Response · Author response to Decision Letter 1]

9 Feb 2024

I have attached a response to specific reviewer and editor comments.

---

## [Editor Report · Decision Letter 2]

27 Feb 2024

PONE-D-23-20697R2Wetland vegetation composition and ecology of Lake Abaya in southern Ethiopia.PLOS ONE

Dear Dr. Gojamme,

Thank you for submitting your manuscript to PLOS ONE. After careful consideration, we feel that it has merit but does not fully meet PLOS ONE’s publication criteria as it currently stands. Therefore, we invite you to submit a revised version of the manuscript that addresses the points raised during the review process.

We look forward to receiving your revised manuscript.

Kind regards,

Dharmendra Kumar Meena

Academic Editor

PLOS ONE

Journal Requirements:

**Additional Editor Comments:**

Minor revision is suggested

---

## [Author Response · Author response to Decision Letter 2]

18 Mar 2024

Response to reviewer's comments submitted as an attachment file were previously uploaded as a rebuttal letter that responds to each point raised by the academic editor and reviewer(s). This letter was uploaded as a separate file labeled 'Response to Reviewers' as shown below. However, I have received similar file as attachment for second time.

I have uploaded the figure files to the Preflight Analysis and Conversion Engine (PACE) digital diagnostic tool, so as to ensure that figures meet PLOS requirements.

I have reviewed the reference list to ensure that it is complete and correct.

I have cross-checked the papers cited in the manuscript against The Retraction Watch Database online and found no retracted article have been cited.

Response to Reviewers

Some typographical or grammatical errors has been corrected at revision to make the article clear, correct, and unambiguous.

Reviewer #1: 

1. The manuscript is generally well written with adequate statistical analysis. However, the discussion section is thin with some results not discussed at all.

To make the discussion section strong, explanation on the relationship of some important factors with plant community types is provided. Discussion of results on plant diversity and distribution is included and some discussions unrelated to results has been avoided to match the number of results presented.

2.The author needs to state how sampling effort was measured.

The total number of plots, and the total area sampled, plot sizes and their intervals (distance) between plots used in different vegetation types were stated clearly and the references cited. 

3. The citation of references is good, but majority are quite old considering the dynamism in wetlands loss/degradation.

Some recent references on the status of wetlands cited.

4. The recommendations and conclusions do not clearly originate from the results perhaps due to the thin discussion. In fact, some appear to be from general knowledge.

The conclusion and recommendation section are updated to match the results and some general facts removed.

5. Most of the comments are included in the manuscript as comments.

The manuscript is edited as per the comments included.

Reviewer #2: To the Editor and the author

Introduction: The often-overlooked functions of wetlands, such as their role in purification and carbon sequestration………

The role of wetlands in water purification and carbon sequestration included in the introduction section. 

Study Area: The graphical representation of the study area depicted raises concerns due to its poor quality. The aspect ratio of latitude and longitude coordinates is not accurately represented, necessitating a revision of the map to ensure an appropriate scale and resolution. 

The map (Figure 1) is revised or drawn again to ensure an appropriate scale and resolution and to accurately represent aspect ratio so as to enhance its quality. The soil type of the study area and some water parameters were briefly described.

Data Collection: The explanation provided for the methods employed in data collection lacks comprehensive detail…….

To provide detail explanation for the methods employed in data collection, further elaborations made. The total number of plots, and the total area sampled, plot sizes and their intervals (distance) between plots used in different vegetation types were stated clearly and the references cited. The table consisting of environmental variables recorded and rated from the study sites is included in the manuscript.

Data Analysis: The utilization of an outdated version of R during the data analysis, when compared to the more recent version 4.3.1, raises concerns…….

The data from which dendrogram was derived is also checked with the more recent version 4.3.1 and similar graphics is produced. The cutoff height was chosen just above 2 based on the researcher’s ecological evaluation or observation in the field, cluster analysis output and the resulting synoptic table, and hence five community types were selected. As mentioned in most literature, researchers can choose to develop a cluster analysis when their main goal is to sort and allocate observations to groups and, from then on, to analyze what the ideal number of clusters formed is based on field observation and results from software analysis.

The application of NMDS (Non-Metric Multidimensional Scaling) facilitates the correlation of environmental factors with the clusters………

Although both methods facilitate the correlation of environmental factors with the clusters or sample plots, the idea is to depict the most accurate representation of relationships of observations/measurements/data points between both categories. CCA sometimes performs well with skewed species distributions, with quantitative noise in species abundance data.

What is the significance of the red cross symbols within Figure 6, representing the Canonical Correspondence Analysis (CCA) involving species and environmental factors? Can these symbols be eliminated if needed?

Species were abbreviated by combining the first four letters from generic names and specific epithets and only higher priority species with high variances are visible and all other less dominant species are indicated by red cross or plus sign (+). 

Response to Academic editors

Journal Requirements:

1.Please ensure that your manuscript meets PLOS ONE's style requirements, including those for file naming

The manuscript is carefully formatted and edited following PLOS ONE style requirements and templates.

2.During our internal evaluation of the manuscript, we found significant text overlap between your submission and previous work…….

I confirm that this manuscript is part of my PhD study work. It has not been published elsewhere and is not under consideration by another journal. Overlapping and duplicating texts in the manuscripts have been rephrased and sources are cited.

3. Thank you for stating the following financial disclosure: …….

I confirm that this manuscript is part of my PhD study work. The author received no specific funding for this work. The amended statement is also included within the cover letter.

4.In your Data Availability statement, you have not specified where the…….

Data Availability statement: Study data is available based up on request. The minimal underlying data is uploaded as supporting information file.

5.We note that [Figure 1] in your submission contain [map/satellite] images which may be copyrighted……… 

Figure 1 is replaced with new map. I confirm that it is taken from my PhD work.

The source of shape file is central statistical agency of Ethiopia; base maps are from Geo-spatial information institution of Ethiopia and Images are from US GEOLOGICAL SURVEY/European space agency/Geo-spatial information institution of Ethiopia.

---

## [Editor Report · Decision Letter 3]

25 Mar 2024

Wetland vegetation composition and ecology of Lake Abaya in southern Ethiopia.

PONE-D-23-20697R3

Dear Dr. Gojamme

We’re pleased to inform you that your manuscript has been judged scientifically suitable for publication and will be formally accepted for publication once it meets all outstanding technical requirements.

Kind regards,

Dharmendra Kumar Meena

Academic Editor

PLOS ONE

Additional Editor Comments (optional):

The article can be accepted now.
---

## [Editor Report · Acceptance letter]

1 Apr 2024

PONE-D-23-20697R3 

PLOS ONE

Dear Dr. Gojamme, 

I'm pleased to inform you that your manuscript has been deemed suitable for publication in PLOS ONE. Congratulations! Your manuscript is now being handed over to our production team.

Kind regards, 

on behalf of

Dr. Dharmendra Kumar Meena 

Academic Editor

PLOS ONE